# Estimation of Weighted Extropy with Focus on Its Use in Reliability Modeling

**DOI:** 10.3390/e26020160

**Published:** 2024-02-11

**Authors:** Muhammed Rasheed Irshad, Krishnakumar Archana, Radhakumari Maya, Maria Longobardi

**Affiliations:** 1Department of Statistics, Cochin University of Science and Technology, Cochin 682 022, Kerala, India; irshadstati@gmail.com (M.R.I.); karchuarchana@gmail.com (K.A.); 2Department of Statistics, Government College for Women, Thiruvananthapuram 695 014, Kerala, India; publicationsofmaya@gmail.com; 3Dipartimento di Biologia, Università degli Studi di Napoli Federico II, 80126 Naples, Italy

**Keywords:** weighted extropy, log kernel estimation, kernel estimation, empirical estimation, simulation

## Abstract

In the literature, estimation of weighted extropy is infrequently considered. In this paper, some non-parametric estimators of weighted extropy are given. The validation and comparison of the estimators are implemented with the help of simulation study and data illustration. The usefulness of the estimators is demonstrated using real data sets.

## 1. Introduction

The concept of extropy and its use has been explored rapidly in the recent years. It measures the uncertainty contained in the probability distributions and is considered as the complimentary dual of entropy introduced in [1]. The entropy measure is shift-independent, that is, it is the same for both *X* and X+b and it cannot be applied in some fields such as neurology. Thus, in [2], the notion of weighted entropy measure was introduced. The authors pointed out that occurrence of an event has an impact on uncertainty in two ways. It presents both quantitative and qualitative information. That is, it initially reveals the probability of an event occurring and later demonstrates its efficacy in achieving qualitative features of a goal. It is important to note that the information obtained when a device fails to operate or a neuron fails to release spikes in a specific time interval differs significantly from the information obtained when such events occur in other equally wide intervals. This is why there is a need, in some cases, to employ a shift-dependent information measure that assigns varying measures to these distributions.The importance of the presence of weighted measures of uncertainty was exhibited in [3].

The concept of extropy for a continuous rv
*X* has been presented and discussed across numerous works in the literature. The differential extropy defined by [4] is
(1)$$J\left(X\right)=-\frac{1}{2}\int_{0}^{+\infty }f_{X}^{2}\left(x\right)dx.$$
One can refer to [5] for the extropy properties of order statistics and record values. The applications of extropy in automatic speech recognition can be found in [6]. Various literature sources have presented a range of extropy measures and their extensions. Analogous to weighted entropy, in [7], the concept of weighted extropy was introduced (WE) in the literature. It is given as
(2)$$J^{w}\left(X\right)=-\frac{1}{2}\int_{0}^{+\infty }xf_{X}^{2}\left(x\right)dx.$$

Variable *x* in the integral emphasizes the weight related to the occurrence of event X=x. Here, it assigns more significance to large values of *X*. In the literature, extropy, its different versions and their applications have been studied by several authors (see, for instance, [8,9,10]). In particular, a unified version of extropy in classical theory and in Dempster–Shafer theory was studied in [11].

There are several papers available in the literature that delve into the estimation of extropy and its various versions. Kernel estimation on the functionals of the density function was proposed in [12]. The optimal bandwidth for kernel density functionals is provided in [13]. In [14], a brief explanation was established on optimal bandwidth estimators of kernel density functionals for contaminated data. In [15], estimators of extropy were proposed, and also its application was worked on by testing uniformity. In [16], the concept of length biased sampling in estimating extropy was approached. Research on non-parametric estimation using dependent data is also well-explored in the literature. Work by [17] explained the recursive and non-recursive kernel estimation of negative cumulative extropy under the α-mixing dependence condition. Recently, in [18], the kernel estimation of the extropy function was discussed using α-mixing-dependent data. Moreover, in [19], the log kernel estimation of extropy was introduced.

Even if there are several works available in the literature related to the estimation of extropy, little has been published on WE and its estimation until now. There are situations in which we are forced to use WE instead of extropy. Unlike extropy, the qualitative characteristics of information are also represented here. In [20], the significance of employing WE as opposed to regular extropy in certain scenarios was demonstrated. There are instances where certain distributions possess identical extropy values but exhibit distinct WE values. In such situations, it becomes necessary to opt for WE. The estimators of WE can also be used in the selection of models in the reliability analysis. Here, we tried to find some estimators for WE and validated it using simulation study and data analysis.

The paper is organized as follows: In Section 2, we introduce the log kernel estimation of WE. In Section 3, an empirical kernel smoothed estimator of WE is given. A simulation study is conducted to evaluate the estimators, and we also compare log kernel to kernel estimators of WE in Section 4. Section 5 is devoted to the real data analysis to examine the proposed estimators. Finally, we conclude the study in Section 6.

## 2. Log Kernel Estimation of Weighted Extropy

In this section, we introduce the concept of log kernel-based estimation of WE.

Let us define an rv
*X* with unknown pdf
$f_{X}\left(x\right)$. We assume that *X* is defined on *R* and $f_{X}\left(x\right)$ is continuously differentiable. We suppose $\left\{X_{i};1\leq i\leq n\right\}$ is a sequence of identically distributed rvs. The most commonly used estimator of $f_{X}\left(x\right)$ is the kernel density estimator (KDE), given by [21,22] as
(3)$${\hat{f}}_{X}\left(x\right)=\frac{1}{nh}\sum_{i=1}^{n}K\left(\frac{x-X_{i}}{h}\right),$$
where K(x) is the kernel function which satisfies the following conditions:$\int_{R}K\left(x\right)dx=1$,$\int_{R}xK\left(x\right)dx=0$,$\int_{R}x^{2}K\left(x\right)dx=1$,$\int_{R}K^{2}\left(x\right)dx$ < +∞.

Here, bandwidth parameter h → 0 and nh → +∞ as *n* → +∞.

When probability density functions are estimated in a non-parametric way, standard KDE is frequently used. However, when we deal with data that fit distributions with heavy tails, multiple modes, or skewness, particularly those with positive values these estimators may lose their effectiveness. In all of these scenarios, applying a transformation, we can yield more consistent results. Such transformation involves a logarithmic transformation to create a non-parametric KDE. An important aspect of the logarithmic transformation is its ability to compress the right tail of the distribution. The obtained KDE are called logarithmic KDE (denoted as L−KDE) (refer to [23]). Let us define Y=log(X), $Y_{i}=log\left(X_{i}\right)$; i=1,2,…,n and let $f_{Y}\left(y\right)$ be the pdf of *Y*. The L−KDE is defined as
(4)$$\begin{array}{cc} {\hat{f}}_{log}\left(x\right) & =\frac{1}{nh}\sum_{i=1}^{n}\frac{1}{x}K\left(\frac{logx-logX_{i}}{h}\right)\\ \; & =\frac{1}{n}\sum_{i=1}^{n}L\left(x,X_{i},h\right), \end{array}$$
where $L\left(x,z,h\right)=\frac{1}{xh}K\left(log{\left(\frac{x}{z}\right)}^{\frac{1}{n}}\right)$ is the log kernel function with bandwidth h>0 at location parameter *z*. For any z,h∈(0,+∞), L(x,z,h) satisfies conditions L(x,z,h)≥0 for all x∈(0,+∞) and $\int_{0}^{+\infty }L\left(x,z,h\right)dx=1$.

For any X∈(0,+∞),
(5)$$Bias\left({\hat{f}}_{log}\left(x\right)\right)=\frac{h^{2}}{2}\left[f_{X}\left(x\right)+3xf_{X}^{\left(1\right)}\left(x\right)+x^{2}f_{X}^{\left(2\right)}\left(x\right)\right]+o\left(h^{2}\right),$$
(6)$$Var\left({\hat{f}}_{log}\left(x\right)\right)=\frac{C_{k}}{nh}\frac{f_{X}\left(x\right)}{x}+o\left(\frac{1}{nh}\right),$$
where $C_{k}=\int_{R}K^{2}\left(z\right)dz$.

We let $\left(X_{1},X_{2},\ldots ,X_{n}\right)$ be a sample of identically distributed observations. We obtain the L−KDE for WE by using the estimator defined in Equation (Equation 4).

The L−KDE for the WE function is
(7)$${\hat{J}}_{n}^{w}\left(X\right)=-\frac{1}{2}\int_{0}^{+\infty }x{\hat{f}}_{log}^{2}\left(x\right)dx,$$
which again can be alternatively expressed as
(8)$$=-\frac{1}{2}\int_{0}^{+\infty }dy\int_{y}^{+\infty }{\hat{f}}_{log}^{2}\left(x\right)dx.$$
The following theorem gives the expression for bias and variance of the L−KDE of WE.

**Theorem** **1.**
*Assume that the conditions given in Section 2 are satisfied in the case of log kernel function L(x) and bandwidth h. Then, the bias and variance of L−KDE
${\hat{J}}_{n}^{w}\left(X\right)$ are given, respectively, as*

(9)
$$Bias\left({\hat{J}}_{n}^{w}\left(X\right)\right)⋍-\int_{0}^{+\infty }dy\int_{y}^{+\infty }\frac{h^{2}}{2}\left[f_{X}\left(x\right)+3xf_{X}^{\left(1\right)}\left(x\right)+x^{2}f_{X}^{\left(2\right)}\left(x\right)\right]f_{X}\left(x\right)dx+o\left(h^{2}\right),$$


(10)
$$Var\left({\hat{J}}_{n}^{w}\left(X\right)\right)⋍\frac{C_{k}}{nh}\int_{0}^{+\infty }dy\int_{y}^{+\infty }\frac{f_{X}^{3}\left(x\right)}{x}dx+o\left(\frac{1}{nh}\right),$$

*where $C_{k}=\int_{R}K^{2}\left(z\right)dz$.*


**Proof.** The proof is omitted as it is similar to [19]. □

The following theorem shows that the proposed estimator is consistent.

**Theorem** **2.**
*${\hat{J}}_{n}^{w}\left(X\right)$ is a consistent estimator of $J^{w}\left(X\right)$, where ${\hat{J}}_{n}^{w}\left(X\right)$ and $J^{w}\left(X\right)$ are defined in Equations (Equation 2) and (Equation 7). Also, let L(x) be the log kernel function and h be the bandwidth which satisfies the conditions given in Section 2. Then, we can say that, as n tends to +∞,*

(11)
$${\hat{J}}_{n}^{w}\left(X\right)=-\frac{1}{2}\int_{0}^{+\infty }dy\int_{y}^{+\infty }{\hat{f}}_{log}^{2}\left(x\right)\overset{p}{\to }-\frac{1}{2}\int_{0}^{+\infty }dy\int_{y}^{+\infty }f_{X}^{2}\left(x\right)dx=J^{w}\left(X\right).$$



**Proof.** Since the proof is similar to that of [19], it is omitted. □

The below theorem shows that the L−KDE of WE is integratedly uniformly consistent in the quadratic mean estimator of $J^{w}\left(X\right)$.

**Theorem** **3.**
*Consider log kernel function L(x) and bandwidth parameter h that fulfills the conditions outlined in Section 2. If ${\hat{J}}_{n}^{w}\left(X\right)$ is L−KDE according to Equation (Equation 7), then ${\hat{J}}_{n}^{w}\left(X\right)$ is integratedly uniformly consistent in the quadratic mean estimator of $J^{w}\left(X\right)$.*


**Proof.** As the proof resembles that of [19], it is omitted here. □

Here, we provide the expression for the optimal bandwidth of ${\hat{J}}_{n}^{w}\left(X\right)$.

### Optimal Bandwidth

Here, we offer the expression for the optimal bandwidth using mean integrated square error (MISE). The MISE of ${\hat{J}}_{n}^{w}\left(X\right)$ is given as
(12)$$MISE\left({\hat{J}}_{n}^{w}\left(X\right)\right)=E\left[\int_{0}^{+\infty }{\left({\hat{J}}_{n}^{w}\left(X\right)-J^{w}\left(X\right)\right)}^{2}\right]dx.$$
Using the expression for bias and variance given in Equations (Equation 9) and (Equation 10), the MISE of ${\hat{J}}_{n}^{w}\left(X\right)$ is given as
(13)$$\begin{array}{cc} MISE({\hat{J}}_{n}^{w}\left(X\right))⋍ & \int_{0}^{+\infty }[{\left(-\frac{h^{2}}{2}\int_{0}^{+\infty }dy\int_{y}^{+\infty }\left[f_{X}\left(x\right)+3xf_{X}^{\left(1\right)}\left(x\right)+x^{2}f_{X}^{\left(2\right)}\left(x\right)\right]f_{X}\left(x\right)dx\right)}^{2}\\ \; & +\frac{C_{k}}{nh}\int_{0}^{+\infty }dy\int_{y}^{+\infty }\frac{f_{X}^{3}\left(x\right)}{x}dx]dx+o\left(h^{4}\right)+o\left(\frac{1}{nh}\right). \end{array}$$
The asymptotic MISE (AMISE) can be obtained by ignoring the higher-order terms and is given as
(14)$$\begin{array}{cc} AMISE= & \frac{h^{4}}{4}\int_{0}^{+\infty }\left[{\left(\int_{0}^{+\infty }dy\int_{y}^{+\infty }\left[f_{X}\left(x\right)+3xf_{X}^{\left(1\right)}\left(x\right)+x^{2}f_{X}^{\left(2\right)}\left(x\right)\right]f_{X}\left(x\right)dx\right)}^{2}\right]dx\\ \; & +\frac{1}{nh}\int_{0}^{\infty }\left[C_{k}\int_{0}^{+\infty }dy\int_{y}^{+\infty }\frac{f_{X}^{3}\left(x\right)}{x}dx\right]dx. \end{array}$$

The optimal bandwidth is then attained after minimizing AMISE with respect to h, and it is given by
$$\begin{array}{cc} h & ={\left(\frac{\int_{0}^{+\infty }\left[C_{k}\int_{0}^{+\infty }dy\int_{y}^{+\infty }\frac{f_{X}^{3}\left(x\right)}{x}dx\right]dx}{\int_{0}^{+\infty }\left[{\left(\int_{0}^{+\infty }dy\int_{y}^{+\infty }\left[f_{X}\left(x\right)+3xf_{X}^{\left(1\right)}\left(x\right)+x^{2}f_{X}^{\left(2\right)}\left(x\right)\right]f_{X}\left(x\right)dx\right)}^{2}\right]dx}\right)}^{\frac{1}{5}}n^{-\frac{1}{5}}\\ \; & =o(n^{-\frac{1}{5}}). \end{array}$$

## 3. Empirical Estimation of Weighted Extropy

Non-parametric estimation is a widely employed technique in various research papers for estimating extropy and its associated measures. One common approach within non-parametric estimation is the use of kernel density estimation, which is a popular method in the literature used in order to obtain smoothed estimates.

In this section, we introduce the empirical method for estimating pdf to assess WE. This estimation is achieved through the utilization of a non-parametric KDE (see [24,25]). The empirical kernel smoothed estimator for WE is
$$\begin{array}{cc} {\hat{J}}_{n1}^{w}\left(X\right) & =-\frac{1}{2}\int_{0}^{+\infty }x{\hat{f}}_{X}^{2}\left(x\right)dx\\ \; & =-\frac{1}{2}\sum_{i=1}^{n-1}\int_{X_{i:n}}^{X_{i+1:n}}x{\hat{f}}_{X}^{2}\left(x\right)dx\\ \; & =-\frac{1}{2}\sum_{i=1}^{n-1}\frac{X_{i:n}^{2}-X_{i+1:n}^{2}}{2}{\hat{f}}_{X}^{2}\left(X_{i:n}\right)\\ \; & =-\frac{1}{4}\sum_{i=1}^{n-1}\left(X_{i:n}^{2}-X_{i+1:n}^{2}\right){\hat{f}}_{X}^{2}\left(X_{i:n}\right), \end{array}$$
where $f_{X}\left(.\right)$ is the KDE given by [21] and $X_{i:n}$ is the $i^{th}$ order statistic of the random sample.

**Example** **1.**
*Let samples $X_{i}^{\prime }s$ be from the distribution with pdf = 2x,0<x<1. Then, $X^{2}$ follows standard uniform distribution. Moreover, $Z_{i+1}=\frac{X_{i:n}^{2}-X_{i+1:n}^{2}}{2}$ is a beta distribution with mean and variance, respectively, as $\frac{1}{2(n+1)}$ and $\frac{n}{4{\left(n+1\right)}^{2}\left(n+2\right)}$. Then, the mean and variance of ${\hat{J}}_{n1}^{w}\left(X\right)$ are given by*

(15)
$$E\left({\hat{J}}_{n1}^{w}\left(X\right)\right)=-\frac{1}{4(n+1)}\sum_{i=1}^{n-1}{\hat{f}}_{X}^{2}\left(X_{i:n}\right),$$

*and*

$$V\left({\hat{J}}_{n1}^{w}\left(X\right)\right)=\frac{n}{16{\left(n+1\right)}^{2}\left(n+2\right)}\sum_{i=1}^{n-1}{\hat{f}}_{X}^{4}\left(X_{i:n}\right),$$

*where ${\hat{f}}_{X}\left(.\right)$ is defined in Equation (Equation 3).*

*Table 1 shows the values of mean and variance of the samples of Example 1. Hence, it is clear that the values of mean is changing and the variance is tending to zero when the sample size increases. It is therefore clear that the mean and variance of empirical estimators are influenced by the size of the sample.*


**Example** **2.**
*Suppose X follows Rayleigh distribution with parameter 1. Then, $X^{2}$ follows exponential distribution and $Z_{i+1}=\frac{X_{i:n}^{2}-X_{i+1:n}^{2}}{2}$ is distributed as exponential distribution with mean = $\frac{1}{2(n-i)}$, for i=1,2,…,n−1. The mean and variance of ${\hat{J}}_{n1}^{w}\left(X\right)$ are*

(16)
$$E\left({\hat{J}}_{n1}^{w}\left(X\right)\right)=-\frac{1}{4}\sum_{i=1}^{n-1}\frac{{\hat{f}}_{X}^{2}\left(X_{i:n}\right)}{n-i},$$


(17)
$$V\left({\hat{J}}_{n1}^{w}\left(X\right)\right)=\frac{1}{16}\sum_{i=1}^{n-1}\frac{{\hat{f}}_{X}^{4}\left(X_{i:n}\right)}{{\left(n-i\right)}^{2}}.$$


*From Table 2, it is clear that the variance is decreasing to zero and the mean is increasing in the case of Rayleigh distribution with parameter one, which indicates the dependence of empirical estimators on sample size.*


## 4. Simulation Study

We manage a simulation study to evaluate the performance of the presented estimators. Random samples are generated corresponding to different sample sizes from some standard distributions, and then both bias and root mean square (RMSE) are calculated for 10,000 samples. Bandwidth parameter h is determined using the plug-in method as proposed in [26].

To enable a comparison between L−KDE and KDE of WE, we again propose a KDE for WE using Equation (Equation 3). The estimator is given by
(18)$$\begin{array}{cc} {\hat{J}}_{nk}^{w}\left(X\right) & =-\frac{1}{2}\int_{0}^{+\infty }x{\hat{f}}_{X}^{2}\left(x\right)dx\\ \; & =-\frac{1}{2}\int_{0}^{+\infty }dy\int_{y}^{+\infty }{\hat{f}}_{X}^{2}\left(x\right)dx, \end{array}$$
where ${\hat{f}}_{X}\left(x\right)$ is the KDE given in [21]. Using the consistency property of the KDE, it is clear that the proposed estimator in Equation (Equation 18) for WE is also consistent. To lay the ground work for comparison, we generate samples from exponential distribution, log normal distribution, a heavy-tailed distribution and uniform distribution. The Gaussian log transformed kernel and the Gaussian kernel are the kernel functions used for simulation.

From the above Table 3, Table 4 and Table 5, it is clear that the RMSE and bias of both estimators are decreasing with sample size. The decreasing RMSE indicates that estimator predictions are approaching the true values with larger sample sizes, demonstrating enhanced accuracy and efficiency in estimation. The decreasing bias also shows the accuracy of the estimators.

The comparison of bias and RMSE between the presented estimators in the simulation for WE reveals that L−KDE slightly outperforms KDE in certain scenarios, particularly when dealing with heavy-tailed distribution and skewed distributions.

## 5. Data Analysis

In this section, we performed a comparison study and validated the accuracy of the proposed estimators using real data analysis. In each of the three scenarios, the bandwidth parameter employed for estimation was derived from the bandwidth proposed in [26].

### 5.1. Data 1

The comparison between L−KDE and KDE of WE was demonstrated using the data given in [27]. The data demonstrate the quantity of thousands of cycles to failure for electrical appliances in a life test.

The graphical representation in Figure 1 indicates the presence of slight skewness in the dataset. We fit exponential distribution with parameter 0.640 to the data. Upon analyzing the Q-Q plot in Figure 2, it becomes evident that the exponential distribution is a suitable model for the observed data. The p-value obtained for the Kolmogorov–Smirnov test (0.124) is 0.390, which reveals that exponential distribution is a good fit to the data. The estimate obtained using maximum likelihood estimation is −0.125.

The estimate of WE earned using log kernel and kernel estimation are ${\hat{J}}_{n}^{w}\left(X\right)=-0.127$, ${\hat{J}}_{nk}^{w}\left(X\right)=-0.144$ and ${\hat{J}}_{n1}^{w}\left(X\right)=-0.148$. Hence, from the closeness of estimates to the maximum likelihood estimate of WE, it is clear that estimator ${\hat{J}}_{n}^{w}\left(X\right)$ performs better than the other two estimators.

### 5.2. Data 2 (Heavy-Tailed Data)

Again, we illustrate the comparison between the three estimators using the data from [28]. The data represent the remission times (months) of 137 cancer patients. A kurtosis value of 15.195 is obtained. It is exceptionally high and suggests a very heavy-tailed or leptokurtic distribution. Hence, log normal distribution is fitted to the data and the parameters obtained are
$$\hat{\mu }=1.756,\;\;\hat{\sigma }=1.066.$$

Figure 3 indicates the presence of rightly skewed heavy-tailed data. Upon examination of the Q-Q plot presented in Figure 4, it is clear that the data align well with the characteristics of log normal distribution, indicating that the log normal model is an appropriate fit for the observed dataset. Using the Kolmogorov–Smirnov test with a statistic of 0.06 and a *p*-value of 0.591, it is clear that log normal distribution is the best fit here.

The estimates of WE using the proposed estimators and by maximum likelihood estimation are calculated for these data. We obtain ${\hat{J}}_{n}^{w}\left(X\right)=-0.1346$, ${\hat{J}}_{nk}^{w}\left(X\right)=-0.1418$, and ${\hat{J}}_{nk}^{w}\left(X\right)=-18.952$. The estimate of WE using maximum likelihood estimation is secured as −0.1323, which signifies that the L−KDE of WE performs better than the WE estimated with standard kernel estimation methods when dealing with heavy-tailed data.

### 5.3. Data 3 (The Time until Failure of the Three Systems)

The data are obtained from [29]. The observations represent three reparable systems observed until the time of their 12th failure. They clarify that the three identically designed systems exhibit distinct behaviors, with their repair rates demonstrating a decreasing trend indicative of improvement in one system, a stable linear trend in another system, and an increasing trend signifying deterioration in the third system. Figure 5 shows the density plot of the three systems. Table 6 shows the value of suggested estimators of WE for these systems.

According to [30], the system or component which is said to have high uncertainty is less reliable. In accordance with this concept, we can infer that System 3 is less reliable than System 1 and System 2 with regard to the three proposed estimators. Using repair rates, in [29], System 3 was also mentioned as the deteriorating system. This example vividly demonstrates how the estimation of WE is useful in choosing a reliable system among the several available competing models.

## 6. Conclusions

In this article, we considered non-parametric estimation of WE. L−KDE and the empirical kernel smoothed estimator for WE were depicted. The bias, variance, optimal bandwidth and some properties of the L−KDE of the extropy function were also established here. KDE was also proposed to enable a comparison with the proposed L−KDE. We ensured the accuracy of the three estimators by evaluating their performance using measures such as bias and RMSE. We determined that in some situations, for example, when dealing with heavy tailed or skewed data sets, the L−KDE of WE performs slightly better than the other two estimators. The real data analyses also involved an assessment of the performance of the estimator and its utility in reliability modeling. We also demonstrated how WE is beneficial when choosing a reliable system from various competing models, highlighting its practicality in the selection process.

## Figures and Tables

**Figure 1 entropy-26-00160-f001:**
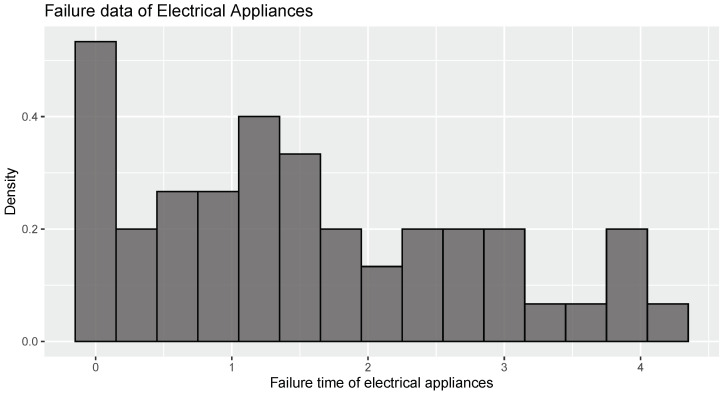
Histogram for “Failure time of Electrical Appliances” data.

**Figure 2 entropy-26-00160-f002:**
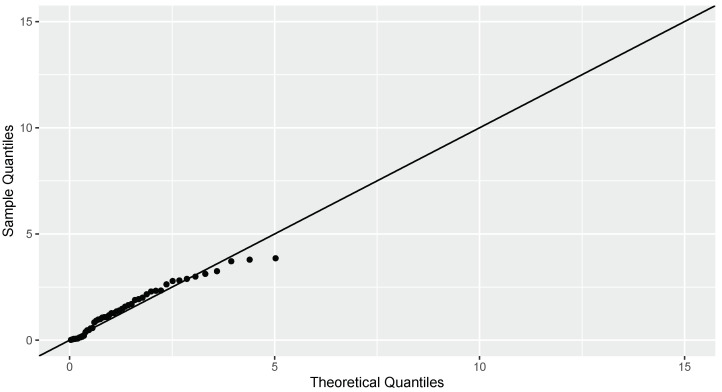
The Q-Q plot depicting the goodness of fit for an exponential distribution.

**Figure 3 entropy-26-00160-f003:**
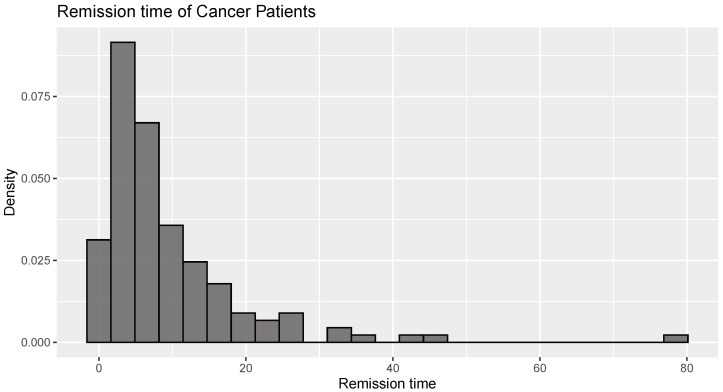
Histogram for “Remission time of cancer patients” data.

**Figure 4 entropy-26-00160-f004:**
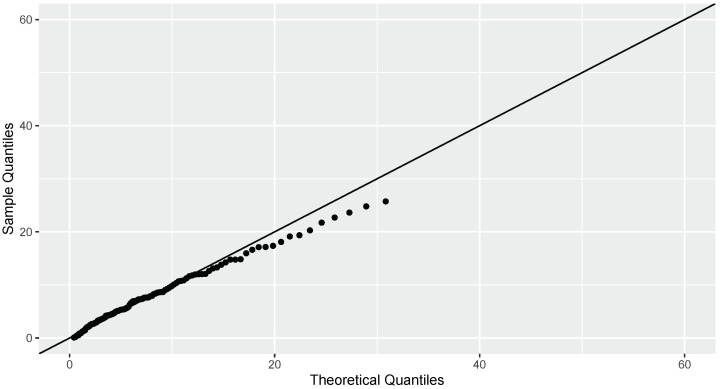
The Q-Q plot depicting the goodness of fit for log normal distribution.

**Figure 5 entropy-26-00160-f005:**
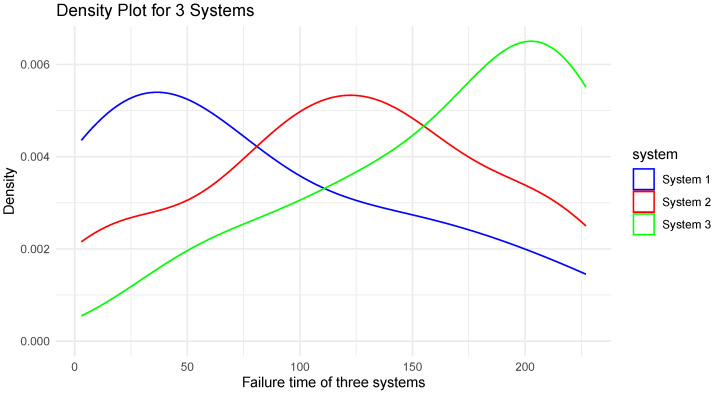
Density plot of “Failure time of three systems”.

**Table 1 entropy-26-00160-t001:** Mean and Variance of ${\hat{J}}_{n1}^{w}\left(X\right)$ for the distribution with pdf = 2x,0<x<1.

*n*	Mean	Variance
10	−0.58602	0.03039
20	−0.57820	0.02182
30	−0.55622	0.01888
40	−0.56181	0.01083
50	−0.52369	0.00777
100	−0.52269	0.00697
500	−0.49937	0.00347

**Table 2 entropy-26-00160-t002:** Mean and Variance of ${\hat{J}}_{n1}^{w}\left(X\right)$ for Rayleigh distribution with parameter = 1.

*n*	Mean	Variance
10	−0.46220	0.02453
20	−0.33956	0.01626
30	−0.29213	0.00173
40	−0.28919	0.00121
50	−0.26677	0.00094
100	−0.25975	0.00024
500	−0.22331	0.00008

**Table 3 entropy-26-00160-t003:** Estimated value (H), |bias| and RMSE of ${\hat{J}}_{n}^{w}\left(X\right)$, ${\hat{J}}_{nk}^{w}\left(X\right)$, and ${\hat{J}}_{n1}^{w}\left(X\right)$ from standard exponential distribution with $J^{w}\left(X\right)=-0.125$.

${\hat{J}}_{n}^{w}\left(X\right)$	*n*	50	100	150	200	250
	H	−0.1183	−0.11886	−0.11904	−0.11909	−0.11978
	|bias|	0.00670	0.00614	0.00596	0.00596	0.00591
	RMSE	0.01760	0.01303	0.01048	0.00948	0.00836
	*n*	300	350	400	450	500
	H	−0.11952	−0.11975	−0.12025	−0.12028	−0.12061
	|bias|	0.00548	0.00525	0.00475	0.00472	0.00439
	RMSE	0.00836	0.00774	0.00707	0.00707	0.00632
${\hat{J}}_{nk}^{w}\left(X\right)$	*n*	50	100	150	200	250
	H	−0.13364	−0.13254	−0.13088	−0.13002	−0.12998
	|bias|	0.00864	0.00754	0.00588	0.00502	0.00498
	RMSE	0.01732	0.01140	0.00948	0.00874	0.00855
	*n*	300	350	400	450	500
	H	−0.12989	−0.12976	−0.12965	−0.12946	−0.12955
	|bias|	0.00489	0.00476	0.00465	0.00446	0.00455
	RMSE	0.00852	0.00832	0.00827	0.00817	0.00807
${\hat{J}}_{n1}^{w}\left(X\right)$	*n*	50	100	150	200	250
	H	−0.16379	−0.14418	−0.13857	−0.13496	−0.13285
	|bias|	0.03879	0.01918	0.01357	0.00996	0.00785
	RMSE	0.05059	0.02280	0.01643	0.01224	0.01
	*n*	300	350	400	450	500
	H	−0.13234	−0.13111	−0.13043	−0.13005	−0.12976
	|bias|	0.00734	0.00611	0.00543	0.00505	0.00476
	RMSE	0.00948	0.00836	0.00824	0.00807	0.00807

**Table 4 entropy-26-00160-t004:** Estimated value (H), |bias| and RMSE of ${\hat{J}}_{n}^{w}\left(X\right)$, ${\hat{J}}_{nk}^{w}\left(X\right)$, and ${\hat{J}}_{n1}^{w}\left(X\right)$ from lognormal distribution with $J^{w}\left(X\right)=-0.14105$.

${\hat{J}}_{n}^{w}\left(X\right)$	*n*	50	100	150	200	250
	H	−0.1437	−0.14243	−0.14199	−0.14199	−0.14189
	|bias|	0.00265	0.00139	0.00095	0.00095	0.00084
	RMSE	0.01581	0.01095	0.00894	0.00894	0.00707
	*n*	300	350	400	450	500
	H	−0.14175	−0.14121	−0.14127	−0.14155	−0.1414
	|bias|	0.0007	0.00016	0.00012	0.00011	0.00006
	RMSE	0.00632	0.00450	0.00447	0.00423	0.00411
${\hat{J}}_{nk}^{w}\left(X\right)$	*n*	50	100	150	200	250
	H	−0.14621	−0.14375	−0.14241	−0.14207	−0.14144
	|bias|	0.00517	0.00271	0.00136	0.00103	0.00039
	RMSE	0.01612	0.01140	0.00836	0.00707	0.00632
	*n*	300	350	400	450	500
	H	−0.14139	−0.14138	−0.14127	−0.14126	−0.14093
	|bias|	0.00037	0.00034	0.00023	0.00022	0.00012
	RMSE	0.00632	0.00547	0.00547	0.00547	0.00547
${\hat{J}}_{n1}^{w}\left(X\right)$	*n*	50	100	150	200	250
	H	−0.223	−0.17574	−0.16491	−0.15942	−0.15744
	|bias|	0.08195	0.03469	0.02386	0.01837	0.01639
	RMSE	0.06103	0.05049	0.03286	0.02738	0.02645
	*n*	300	350	400	450	500
	H	−0.15401	−0.15218	−0.15072	−0.15015	−0.14888
	|bias|	0.01296	0.01113	0.00967	0.00911	0.00783
	RMSE	0.02000	0.01581	0.01414	0.01449	0.01183

**Table 5 entropy-26-00160-t005:** Estimated value (H), |bias| and RMSE of ${\hat{J}}_{n}^{w}\left(X\right)$, ${\hat{J}}_{nk}^{w}\left(X\right)$, and ${\hat{J}}_{n1}^{w}\left(X\right)$ from standard uniform distribution with $J^{w}\left(X\right)=-0.25$.

${\hat{J}}_{n}^{w}\left(X\right)$	*n*	50	100	150	200	250
	H	−0.2097	−0.21562	−0.21826	−0.22059	−0.22285
	|bias|	0.04030	0.03438	0.03174	0.02941	0.02715
	RMSE	0.05347	0.04289	0.03781	0.03464	0.03146
	*n*	300	350	400	450	500
	H	−0.22277	−0.22426	−0.22511	−0.22575	−0.22668
	|bias|	0.02723	0.02574	0.02489	0.02425	0.02332
	RMSE	0.03114	0.02966	0.02810	0.02756	0.02607
${\hat{J}}_{nk}^{w}\left(X\right)$	*n*	50	100	150	200	250
	H	−0.22576	−0.22786	−0.22829	−0.23056	−0.23045
	|bias|	0.02424	0.02214	0.02171	0.01955	0.01944
	RMSE	0.04123	0.03162	0.02828	0.02588	0.02569
	*n*	300	350	400	450	500
	H	−0.23201	−0.23276	−0.23325	−0.23399	−0.23379
	|bias|	0.01799	0.01724	0.01675	0.01621	0.01601
	RMSE	0.02302	0.02167	0.02097	0.01974	0.01974
${\hat{J}}_{n1}^{w}\left(X\right)$	*n*	50	100	150	200	250
	H	−0.22669	−0.22713	−0.22892	−0.23084	−0.23194
	|bias|	0.02331	0.02287	0.02108	0.01916	0.01806
	RMSE	0.04147	0.03271	0.02915	0.02569	0.02366
	*n*	300	350	400	450	500
	H	−0.23168	−0.23195	−0.23317	−0.23335	−0.23361
	|bias|	0.01832	0.01805	0.01683	0.01665	0.01639
	RMSE	0.02345	0.02213	0.02121	0.02024	0.01974

**Table 6 entropy-26-00160-t006:** Values of ${\hat{J}}_{n}^{w}\left(X\right)$, ${\hat{J}}_{nk}^{w}\left(X\right)$ and ${\hat{J}}_{n1}^{w}\left(X\right)$ for the three systems.

	${\hat{J}}_{n}^{w}\left(X\right)$	${\hat{J}}_{\mathrm{nk}}^{w}\left(X\right)$	${\hat{J}}_{n1}^{w}\left(X\right)$
System 1	−0.09638	−0.12426	−0.14345
System 2	−0.19953	−0.20431	−0.19666
System 3	−0.39227	−0.41138	−0.30690

## Data Availability

The data utilized in this study is taken from published articles, appropriately cited and listed in the references.

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
