# Peer review of "Estimation of Weighted Extropy with Focus on Its Use in Reliability Modeling"

_entropy, 2024, doi:10.3390/e26020160_

Round 1

Reviewer 1 Report

Comments and Suggestions for Authors

The clarity and readability of the paper could be improved by taking the following points into consideration:

1. Page 2, line 38. The X_i are described as a sequence of iid random variables. In fact these are samples from a common distribution.

2. In equation (6), h appears without index n, while in the text below it appears as h_n. The text references to h_n while (6) contains h.

3. Page 3, line 83: "we have introduced" should be "we introduce". 

4. Page 5, line 107: "integratedly" is not a word.

5. Page 1, lines 15-16: the sentence "Being quantitative and qualitative..." is unclear. what is "it"? Please reformulate.

Comments on the Quality of English Language

The language can be improved at a few places, as shown in the more detailed comments above.

Author Response

Dear Reviewer,

we have done all the corrections you have required. Thank you very much for your useful comments

Reviewer 2 Report

Comments and Suggestions for Authors

Author Response

Dear Reviewer,

we have done all the corrections you have required. Thank you very much for your useful comments.

Round 2

Reviewer 2 Report

Comments and Suggestions for Authors

Please see the attached comments.
